# Mesenchymal Stromal Cells Derived from Dental Tissues: Immunomodulatory Properties and Clinical Potential

**DOI:** 10.3390/ijms25041986

**Published:** 2024-02-06

**Authors:** Luis Ignacio Poblano-Pérez, Marta Elena Castro-Manrreza, Patricia González-Alva, Guadalupe R. Fajardo-Orduña, Juan José Montesinos

**Affiliations:** 1Mesenchymal Stem Cell Laboratory, Oncology Research Unit, Oncology Hospital, National Medical Center (IMSS), Mexico City 06720, Mexico; poblanoperez@hotmail.com (L.I.P.-P.); guadalupefajardo@hotmail.com (G.R.F.-O.); 2Immunology and Stem Cells Laboratory, FES Zaragoza, National Autonomous University of Mexico (UNAM), Mexico City 09230, Mexico; elmar_ca@yahoo.com.mx; 3Tissue Bioengineering Laboratory, Postgraduate Studies, Research Division, Faculty of Dentistry, National Autonomous University of Mexico (UNAM), Mexico City 04510, Mexico; goap.unam@gmail.com

**Keywords:** mesenchymal stem/stromal cells, dental tissue, immunoregulation, immune cells, immunoregulatory mechanisms

## Abstract

Mesenchymal stem/stromal cells (MSCs) are multipotent cells located in different areas of the human body. The oral cavity is considered a potential source of MSCs because they have been identified in several dental tissues (D-MSCs). Clinical trials in which cells from these sources were used have shown that they are effective and safe as treatments for tissue regeneration. Importantly, immunoregulatory capacity has been observed in all of these populations; however, this function may vary among the different types of MSCs. Since this property is of clinical interest for cell therapy protocols, it is relevant to analyze the differences in immunoregulatory capacity, as well as the mechanisms used by each type of MSC. Interestingly, D-MSCs are the most suitable source for regenerating mineralized tissues in the oral region. Furthermore, the clinical potential of D-MSCs is supported due to their adequate capacity for proliferation, migration, and differentiation. There is also evidence for their potential application in protocols against autoimmune diseases and other inflammatory conditions due to their immunosuppressive capacity. Therefore, in this review, the immunoregulatory mechanisms identified at the preclinical level in combination with the different types of MSCs found in dental tissues are described, in addition to a description of the clinical trials in which MSCs from these sources have been applied.

## 1. Introduction

Mesenchymal stem/stromal cells (MSCs) were first discovered in the bone marrow (BM-MSCs) of guinea pigs more than 50 years ago by Friedenstein and his collaborators [1]. The International Society for Cell Therapy defines these cells as adherent cells with a fibroblast morphology that simultaneously express the immunophenotypic markers CD105, CD90, and CD73, present low levels of human leukocyte antigen (HLA) class I molecules and lack HLA class II molecules and markers of hematopoietic and endothelial cells. More recently, new surface molecules have been discovered, and MSCs are now recognized as cells that express STRO-1, CD106, and CD146. Finally, MSCs can differentiate when cultured in a specific inducing medium, at least in adipocytes, chondroblasts, and osteoblasts [2].

BM-MSCs are characterized by regenerative and immunomodulatory properties that can be exploited in clinical treatments for various immune diseases [3,4]. Unfortunately, obtaining them from this source presents drawbacks that reduce their feasibility for use in cell therapy as a result of the painful and invasive process involved [5]. The biological potential and number of cells may vary or decrease with the sex and age of the donor [6,7,8,9,10,11]. When cultivated in the long term, proliferation is negatively affected [12], as are the morphology, differentiation capacity, and genetic stability [13]. Finally, the anti-inflammatory potential of these cells may be compromised when they are obtained from donors with some pathological conditions, such as rheumatoid arthritis [14], which would reduce the possibility of using autologous samples. For these reasons, alternative sources to BM where these inconveniences are minimized are needed.

Currently, MSCs can be obtained from neonatal tissues, including umbilical cord blood (UCB) [15], the umbilical cord (UC), and the placenta [16], as well as adult sources, such as adipose tissue (AT) [17], synovial fluid [18], the skin [19], the lungs [20], the liver [21], peripheral blood [22,23,24,25], and dental tissues [26]. Most MSCs from dental tissues, possess similar properties to MSCs from BM; however, there are still some gaps in the knowledge of their biological characteristics.

The aim of this review is therefore to analyze the immunoregulatory mechanisms of the different types of MSCs found in dental tissues, as well as to describe the preclinical and clinical trials in which MSCs from these sources have been applied. In order to find alternative sources of MSCs to bone marrow that may be applicable in diseases where the immune system is involved, it is necessary to understand the immunological characteristics and mechanisms of D-MSCs.

## 2. MSCs from Dental and Periodontal Tissues

The teeth are essential multifunctional appendages for functions such as speaking or eating [27]. They are divided into two main regions: the upper part or crown and the root, which is anchored within the mandibular and maxillary bones. In humans, there are two sets of teeth: the initial deciduous or primary teeth and the successive permanent (secondary) teeth [28].

Structurally, teeth are made up of three highly mineralized tissues called enamel, dentin, and cement, whose functions include providing support, size, shape, and anchorage [29,30]. Enamel and dentin also protect a fourth nonmineralized tissue known as the dental pulp. This area serves as a reservoir for fibroblasts and is characterized by a vascular–nervous system that features a unique combination of blood vessels, nerves, odontoblasts, and an extracellular matrix [31,32]. The vascular–nervous system plays a fundamental role in tooth function, feeding it from the apex without accessory vascularization. Unlike other oral tissues adjacent to the tooth, this mechanism maintains homeostasis and provides the tooth with neurosensory function, aiding in repair processes [27,33,34]. On the other hand, teeth also rely on other tissues, including the periodontal ligament, the alveolar bone, and the gingival tissue; together, these tissues are known as periodontal tissues or simply the periodontium.

The functions of the periodontium include supporting the tooth and protecting it against invasion by oral microorganisms; it is essential in the immune response and allows the attachment of the tooth to the bone [35,36].

MSCs have been isolated from various anatomical regions of the oral cavity (Figure 1), including the dental pulp (DPSCs) [37,38], periodontal ligament (PDLSCs) [39,40], gingival tissue (GMSCs) [41,42], apical papilla (SCAPs) [43,44], dental follicle (DFSCs) [45,46], human exfoliated deciduous teeth (SHED) [47,48], alveolar bone (ABMSCs), and tooth germ (TGPCs) [34], and these are collectively defined as dental MSCs (D-MSCs) [49]. 

Dental tissues are frequently obtained during routine dental extraction procedures. The dental pieces and/or tissues are kept in phosphate-buffered solutions, usually supplemented with antibiotics and antifungals, and a series of washes with fresh solution are performed to remove unwanted tissues. The tissues of interest are stored at 4 °C before beginning the procedures to obtain MSCs. To access and obtain the dental pulp, the teeth are crushed or the crowns are cut. Other tissues such as the ligament, gingiva, and apical papilla can be separated mechanically by using forceps [50,51,52,53,54,55].

The most common methods used to obtain MSCs are explant and enzymatic digestion. In the first method, the tissues obtained are crushed into small pieces of 1 mm and placed on plates with ideal culture medium for the growth of these cells supplemented with fetal bovine serum and antibiotics. They are then incubated at 37 °C and 5% CO_2_. Additionally, in some procedures, a coverslip can be added over the explants to prevent the tissues from moving during the culture time. On the other hand, enzymatic digestion is carried out through the incubation of the crushed tissues in collagenase I and dispase II. The incubation time can vary from 30 min to 2 h at 37 °C. Once digestion has been carried out, the cells are seeded on plates with the same culture conditions mentioned above. Once the cells adhere, the medium and excess tissues are removed and fresh medium is added to continue cell expansion, and subsequently, the capacity to generate colonies, morphology, immunophenotype, and the differentiation potential characteristic of the MSCs are evaluated [50,51,52,53,54,55]. Although most protocols use enzymatic digestion to obtain D-MSCs, both methods are reliable. They can obtain a high number of cells and there is no solid evidence that either affects the biological properties of the cells. However, isolation methods continue to improve with the purpose of obtaining the greatest number of cells with the necessary quality required to be able to use them in preclinical and clinical research.

D-MSCs have several similar characteristics which are summarized in Table 1; for example, they come from easily accessible tissues, they exhibit fibroblast morphology, they provide a high number of MSCs with high proliferation rates, and, unlike MSCs from other sources, they maintain their qualities for more passages [49,56,57,58,59]. However, differences have been observed in other characteristics, such as the immunophenotype, where the expression of some markers may vary. In addition, although they have similar multipotentialities for differentiation toward chondrocytes and osteoblasts, their adipogenic capacity is lower than that of other tissues, such as BM [58,60,61,62,63,64]. Moreover, D-MSCs derived from neural crest cells possess an increased potential for differentiation into neural cells [65,66,67,68]. Taken together, these findings indicate that although all of these sources originate in related anatomical regions, the biological properties of each can vary, and of particular importance is their immunosuppressive capacity due to their relevance in clinical protocols. In the case of D-MSCs, they have great therapeutic potential for applications in tissue repair, including that of bone, dental, and soft tissues.

## 3. Immunomodulatory Properties of MSCs

In vitro, preclinical and clinical studies have shown that MSCs are capable of regulating inflammatory processes, an essential event for reducing tissue damage and promoting tissue repair [79]. To do this, they migrate to injured sites through the support of adhesion molecules, chemokines, and their receptors, as well as chemoattractant molecules, such as hepatocyte growth factor (HGF), vascular endothelial growth factor (VEGF), transforming growth factor β (TGF-β) 1, granulocyte colony stimulating factor, and tumor necrosis factor α (TNF-α) [80,81,82,83]. Once they reach the site of inflammation, MSCs regulate the proliferation, differentiation, maturation, and production of soluble factors and the cytotoxicity of immune cells [84]. The regulatory mechanisms of MSCs include the secretion of soluble factors, the production of metabolites, cell–cell contact, and the release of extracellular vesicles (EVs).

Some of the anti-inflammatory molecules that MSCs secrete include interleukin (IL)-10 [85,86,87,88], IL-6 [89,90], TGF-β [91,92,93], prostaglandin E2 (PGE2) [94,95,96], gene-6 stimulated by tumor necrosis factor (TSG-6) [97,98], HLA-G5 [99,100], galectins [101], and chemokine ligand 2 with a CC motif (CCL2) [101]. The participation of each secreted factor in the regulation of the immune response will be explained in the following sections.

On the other hand, MSCs express intra- and extracellular enzymes that help the formation of anti-inflammatory metabolites, such as inducible nitric oxide synthase (iNOS), which allows for the generation of nitric oxide (NO) and reduces the migration of immune cells by decreasing chemoattractant molecules and the production of inflammatory cytokines [102,103,104], or indoleamine 2,3-dioxygenase (IDO), which depletes tryptophan in immune cells and generates anti-inflammatory molecules, including kinurenine and picolinic acid [105,106,107]. Finally, MSCs express the ectonucleotidases CD39 and CD73, which generate adenosine (ADO), modulating the production of inflammatory cytokines, cytotoxicity, apoptosis, and the proliferation of immune cells [108,109,110,111,112].

Among their membrane molecules, MSCs express programmed death ligands (PDLs) 1 and 2 [113,114], HLA-G1 [100], cytotoxic T lymphocyte antigen 4 (CTLA-4) [115], and intercellular adhesion molecule-1 (ICAM-1) [116]. Finally, the EVs secreted by MSCs transport and express immunomodulatory molecules and cytokines, such as microRNAs, TGF-β, galectin 1, and PDL-1 [117].

The immunomodulatory capacity of MSCs is of great relevance to the application of these cells in the clinical field. It is necessary to know in depth the biological mechanisms that allow them to sense their microenvironment to activate this property; Preclinical and clinical trials have given good results, and MSCs have been evaluated to improve adverse physiological conditions related to the immune system, which has created great expectations in cell therapy procedures.

## 4. Immunomodulatory Properties of D-MSCs

The anti-inflammatory characteristics of MSCs have been widely described, especially those of MSCs derived from BM, AT, and UCB [118,119]. However, recent studies have shown that D-MSCs also express immunoregulatory molecules (Table 2; Figure 2). In this regard, our workgroup has shown that, in a similar way to that of BM-MSCs, DPSCs, PDLSCs, and GMSCs reduce the proliferation of activated CD3^+^ T lymphocytes and the production of TNF-α and increase the production of anti-inflammatory molecules such as IL-10 and PGE2, in addition to generating regulatory CD4^+^ CD25^+^ Foxp3^+^ T lymphocytes. In these cocultures, some mechanisms of cell contact have also been observed, since, in MSCs, they increase the expression of PDL-1, while in T lymphocytes, the presence of CTLA-4 increases [60]. However, other mechanisms have been described by different workgroups [120]. Table 2 shows the immunoregulatory molecules that have been reported to be expressed by MSCs from different sources, the effects of which on immune cells are described below.

As discussed below, dental tissues are very promising alternative sources of MSCs to bone marrow because such tissues are usually discarded in routine dental procedures and can be obtained without significant ethical concerns. The following subsections indicate the immunomodulatory properties of the different subtypes of D-MSCs.

### 4.1. Dental Pulp MSCs

DPSCs are characterized by a high proliferation rate and immunomodulatory effects that are exerted through different mechanisms that affect different populations of immune cells [173,174,175]. These MSCs secrete IL-6, TGF-β, IDO, HLAG, and HGF, express PDL-1, rely on the Fas-FasL pathway, produce ADO, release EVs to reduce the proliferation of T lymphocytes, induce apoptosis, decrease the differentiation capacity of some subpopulations of proinflammatory T helper (Th) cells, such as Th17 lymphocytes, decrease the secretion of TNF-α and IL-17, and promote the generation of Tregs and the secretion of IL-10 and TGF-β [114,138,139,140,144].

On the other hand, it has been observed that hypoxia-inducible factor 1 (HIF-1) induces DPSCs to decrease the maturation of monocytes toward dendritic cells (DCs) and induce the polarization of M2 macrophages [141]. In addition, they secrete osteoprotegerin and release EVs loaded with microRNAs to inhibit the osteoclastogenesis of myeloid cells and polarize them toward M2 macrophages [142,143]. It has been shown that these types of MSCs also affect natural killer (NK) lymphocytes, since they inhibit their proliferation, induce apoptosis, and decrease their cytotoxicity and the expression of activator molecules such as NKG2D. In addition, MSCs induce the expression of CD73 in NK cells, which favors the generation of ADO [176]. In a recent study, DPSCs cultured with activated T lymphocytes from patients infected with COVID-19 decreased the secretion of several proinflammatory cytokines, including interferon γ (IFN-γ), TNF-α, IL-2, IL-5, IL-9, IL-12 (p70), IL-17A, IL-18, IL-21, IL-23, and IL-2 [177].

On the other hand, DPSCs—or EVs secreted from them—have been used in in vivo studies of induced arthritis in rodents and have been shown to be capable of reducing cartilage and bone tissue wear, the presence of TNF-α and IFN-γ, the production of metalloproteinases, and the differentiation of Th17 cells [178,179,180]. Other studies have shown that they are capable of inhibiting the activation and proliferation of microglia and, thus, reducing the production of proinflammatory cytokines in neuroinflammation caused by spinal cord injury [181]. In this regard, progenitors of DPSCs migrate from the cranial neural crest into the pharyngeal arches, and their genetic line has been shown to have a close relationship with pericytes and glial cells [182].

Moreover, a previously conducted study demonstrated that exosomes derived from lipopolysaccharide (LPS)-preconditioned DPSCs could enhance Shawn cells’ proliferation, migration, and odontogenic differentiation. Different research groups have demonstrated that stimulation of MSCs with different TLR ligands produces proinflammatory mediators, including IL-6, IL-8, and monocyte chemoattractant protein-1 (MCP-1) [183]. These findings suggest that MSCs are involved in the progression of different inflammatory diseases. However, the precise role of MSCs in oral inflammatory conditions—particularly pulpitis and periodontitis—needs further investigation. An interesting observation is that senescent DPSCs can retain an active cellular metabolism and produce functional exosomes, which can penetrate the blood barrier. This property makes exosomes useful therapeutic agents for treating various diseases, including neurological disorders, cancer, and pulmonary diseases, as well as other pathologies [184].

Finally, previously published research has demonstrated that the DPSC-derived extracellular matrix has the potential to enhance the integration of bone grafts and bone regeneration and repair. Hence, it can serve as a potential tool for enhancing the efficacy of various biomaterials (scaffolds and hydrogels) within the field of bone tissue engineering [185].

### 4.2. Periodontal Ligament MSCs

PDLSCs reduce the proliferation of T lymphocytes through the secretion of TGF-β, IDO, HGF, and PGE2 [145,146,147,148]. In addition, PDLSCs release EVs loaded with microRNAs and restore the balance between Tregs and Th17 lymphocytes [186]. On the other hand, the coculture of PDLSCs with DCs reduced the expression of the CD1b glycoprotein in DCs, preventing them from activating T lymphocytes appropriately [187].

PDLSCs modulate the proliferation, differentiation, migration, and production of immunoglobulins in B cells through programmed cell death protein 1 and its ligands PDL-1 and 2. In addition, the transplantation of allogenic human PDLSCs successfully suppressed the progression of bone resorption in a minipig periodontitis model [188]. Furthermore, through the secretion of the chemokines RANTES, eotaxin, IFN-γ, inducible protein 10, and MCP-1, IL-6, IL-8, and IL-1ra promoted the recruitment of neutrophils to sites of infection so that they could carry out their antimicrobial functions [149].

Several in vivo studies have shown that the administration of PDLSCs favors periodontal regeneration by inducing the polarization of M2 macrophages [189] and inhibiting the proliferation and infiltration of T lymphocytes in affected tissues [146,188,190]. On the other hand, the conditioned media (CMs) of these cells contained VEGF, growth factors (insulin-like growth factor binding protein 6), and cytokines (TGF-β 1, 2, and 3; HGF, IL-6, and MCP-1), which were related to their immunomodulatory functions and tissue regeneration. The use of CMs in vitro induced the polarization of M2 macrophages [191], while their administration in rats with periodontal defects induced the regeneration of bone tissue, decreased the presence of IL-1β, IL-6, and TNF-α and increased IL-10 [192,193].

Thorough research has proven that MSCs regulate the immune response during inflammation by producing multiple factors. This production is usually upregulated by inflammatory cytokines and TLR ligands [194,195]. PDLSCs participate in alveolar bone metabolism and cementogenesis, which are processes that are vital for maintaining periodontal tissues. Considering the importance of PDLSCs, the potential effect of LPS on the osteogenic differentiation of PDLSCs and TLR receptors—mainly TLR-4’s capacity to recognize LPS—has been extensively investigated [194,196,197].

The overall results suggest that the impact of LPS on PDLSC differentiation into bone cells depends on the concentration and, to a lesser extent, the source of LPS. Understanding such mechanisms is crucial in identifying the causes of periodontitis and will contribute to the development of effective treatments [198].

### 4.3. Gingival Tissue MSCs

Gingival tissue is the most accessible source of MSCs, and cells are easily isolated and have great potential for ex vivo expansion [198]. In addition to their high capacity for renewal and proliferation, these cells have been shown to have immunomodulatory properties [42]; in vitro, they decrease the proliferation and differentiation of Th17 lymphocytes through the production of PGE2, IDO, iNOS, and IL-10 and the Fas-FasL pathway [42,150,156], and they induce the polarization of M2 macrophages [159].

The immunomodulatory and protective capacities of these cells have been evaluated in animal models of immune pathologies, including experimental colitis [42], induced rheumatoid arthritis [111,152], type 1 diabetes mellitus [153], atherosclerosis [110], graft versus host disease [59,154], lupus nephritis [155], and chemotherapy-induced mucositis [141]. In these models, GMSCs migrate to inflamed regions and lymph nodes near these sites, where they decrease the proliferation and differentiation of Th1, Th2, and Th17 lymphocytes, as well as their ability to infiltrate the affected tissues; in addition, they affect the cytotoxicity of CD8^+^ T cells. On the other hand, they inhibit the proliferation, differentiation, maturation, and activation of B lymphocytes, decrease the migration of DCs, promote the polarization of macrophages to the M2 phenotype, inhibit osteoclastogenesis, and reduce the levels of TNF-α, IFN-γ, IL-17, and IL-4. Some of the most important immunomodulatory mechanisms of these MSCs are the adenosynergic pathway and the production of IL-10 and IDO.

In addition, the administration of GMSCs mitigated the oxidative stress-induced apoptosis of epithelial cells by regulating the activity of manganese superoxide dismutase and hypoxia-inducible factors 1 and 2α [141].

Interestingly, GMSCs induce the expression of CD39 in Tregs, which increases their anti-inflammatory capacity [59]. Additionally, GMSCs have been reported to maintain their proliferative and immunomodulatory potential longer than BM and AT MSCs [59].

In addition, GMSCs release EVs loaded with the IL-1 receptor antagonist (IL-1RA) miR1260b, and they express CD73. These vesicles induce M2 macrophage polarization, improve wound healing, and decrease osteoclastogenesis [151,157,158,159]. Likewise, the biomolecules present in the CMs of these cells decrease inflammation and improve bone regeneration in rats with periodontitis [192].

### 4.4. MSCs of the Apical Papilla

SCAPs have high proliferative potential and self-renewal capacity and low immunogenicity [43,199,200,201]. In vitro, they have been shown to decrease the proliferation of T lymphocytes and induce the generation of Tregs [200,202,203]. The mechanisms and immunomodulatory molecules used by these MSCs have not yet been defined; however, analysis of their secretome revealed the presence of IL-6, TGF-β 1 and 2, and galectin 1, among others [160,161,162,163].

These MSCs have been cocultivated with tissues derived from the spinal cord, and their ability to modulate microglia and mitigate neuroinflammation by increasing arginase 1 and decreasing TNF-α and NOS2, which translates into a neuroprotective effect, has been demonstrated [204,205].

In some in vivo models of experimental colitis, it has been shown that SCAPs can decrease the production of IL-1β, IL-6, and TNF-α and increase the generation of Treg lymphocytes in the lymph nodes. The Fas-FasL pathway seems to be one of the immunomodulatory mechanisms of these MSCs [206]. On the other hand, they also secrete EVs [207], which have been shown to be capable of generating Treg lymphocytes to decrease inflammation in rat models of periodontal bone defects [208].

### 4.5. Dental Follicle MSCs

DFSCs can be isolated from the dental follicle, a loose connective tissue around a developing tooth. They are a group of cells that play a fundamental role in the development and maintenance of the periodontal tissues during tooth development; they are accessible and abundant, have high proliferative and self-renewal potential, and have immunomodulatory properties [209]. DFSCs inhibit the expression of costimulatory molecules in monocytes, as well as the proliferation and differentiation of Th2 lymphocytes, but they allow the generation of Tregs through the secretion of TGF-β and IDO [164,210].

Chen et al. [165] stated that the CMs of these MSCs contained 42 paracrine factors related to their immunomodulatory capacity, of which TGF-β3 and thrombospondin 1 were the most abundant, and the use of these supernatants in macrophage cultures could polarize them to the M2 phenotype. On the other hand, the administration of DFSCs or their CMs to a rat model of acute lung damage similarly decreased the presence of inflammatory cytokines such as MCP-1, IL-1, IL-6, and TNF-α in bronchoalveolar lavage fluid. In addition, they promoted the secretion of IL-10 and the differentiation of M2 macrophages [165].

In a model of induced pulpitis in rats, CMs administration downregulated the ERK1/2 and NF-κB signaling pathways in active cells, which resulted in the suppression of IL-1β, IL-6, and TNF-α and promoted the expression of IL-4 and TGF-β; additionally, CM administration reduced the infiltration of immune cells into the tissues [166]. On the other hand, in a model of myasthenia gravis, its ability to decrease the proliferation of cells in the lymph nodes was observed, as was the production of IL-6, IL-12, and IgG antibodies [211].

### 4.6. MSCs from Human Exfoliated Deciduous Teeth

SHED are cells that have been shown to have different immunomodulatory mechanisms [212]; they reduce the secretion of IL-2, TNF-α, and IFN-γ and increase the secretion of IL-10 when they are cocultured with activated mononuclear cells (MNCs). In addition, they negatively affect the maturation of DCs or induce them to adopt a regulatory profile, which, in turn, generates Treg lymphocytes [167], inhibits the differentiation of Th17 lymphocytes [213], and polarizes macrophages toward the M2 profile [214]. On the other hand, SHED secrete EVs that carry miRNAs. Cultures of these EVs with cartilage cells decreased the production of IL-6, IL-8, and various metalloproteinases [168].

The administration of SHED in models of immune diseases, such as ovariectomy-induced osteoporosis [170], experimental autoimmune encephalitis [215], and induced arthritis [216], decreased the infiltration of T lymphocytes into tissues, the differentiation of Th1 and Th17 cells, and the presence of IFN-γ, TNF-α, IL-1β, and IL-4, which also induced the generation of Tregs. In an experimental periodontitis model, these cells were able to regenerate periodontal tissues, and this effect was attributed to their ability to polarize macrophages toward M2 [214]. Through the secretion of soluble PDL-1, these cells restored the balance between Th17 and Treg lymphocytes in a mouse model of Sjögren’s syndrome [169].

SHED-CMs also possess anti-inflammatory and protective properties; their administration in a model of multiple sclerosis decreased neuronal damage by reducing the production of inflammatory cytokines and inhibiting the infiltration and proliferation of CD4^+^ T lymphocytes, in addition to inducing the polarization of M2 macrophages [217]. In addition, a model of non-alcoholic steatohepatitis protected the liver and intestine through similar mechanisms [218], and it was evidenced that the protective effect of SHED-CMs was because they contained TGF-β and IL-10 [219].

EVs released by SHED have also been used in vivo. In a mouse model of lupus erythematosus, the mRNAs carried by these vesicles reached BM-MSCs, which showed a decrease in telomerase activity and, therefore, the loss of certain biological properties. With the support of EVs, these MSCs recovered their immunomodulatory capacity and, thus, managed to decrease the percentage of Th17 cells and increase the number of Tregs [220]. On the other hand, in a periodontitis model, the use of SHED-derived EVs decreased the presence of IL-6 and TNF-α in inflamed areas and improved bone regeneration [221].

### 4.7. MSCs from Alveolar Bone

ABMSCs were isolated for the first time by Matsubara and collaborators in 2005. These cells have the morphology, immunophenotype, and differentiation capacity of MSCs [222], and they have demonstrated the capacity for bone regeneration in vivo [223,224,225]; however, their immunomodulatory mechanisms have not been studied in depth. To date, it is only known that they inhibit the proliferation of T lymphocytes and monocytes and that they induce the differentiation of the latter towards an M2 phenotype. Its mechanisms are possibly due to its ability to produce IL-6, osteoprotegerin, and tissue inhibitors of metalloproteinase (TIMP)-1 and 2 [171,172].

## 5. The Induction of the Immunoregulatory Capacity of D-MSCs and the Release of Extracellular Vesicles

The importance of an inflammatory environment in the activation of MSCs has been demonstrated in numerous studies. This event allows their different immunosuppressive mechanisms to increase or be induced and is a process mediated mainly by IFN-γ and TNF-α [117], whose effect has been analyzed mainly on BM-MSCs. Regarding MSCs derived from dental tissues, it has been seen that the activation of PDLSCs with IL-1β, TNF-α, or IFN-γ increases the expression of the IDO gene at the protein level; the most pronounced effect is observed with IFN-γ, while stimulation with TNF-α or IFN-γ increases PDL1 and 2 expression at almost the same levels. However, only MSCs pre-treated with IFN-γ or IL-1β are able to affect T lymphocyte proliferation, whereas those treated with TNF-α are not [226]. The same working group observed that PDLSCs stimulated with IL-1β or IFN-γ increase IDO expression for at least 5 days, even after withdrawal of the stimulus. Interestingly, the increased IDO expression over a long period of time is observed only when PDLSCs are stimulated with IFN-γ, whereas the opposite effect is observed in MSCs treated with TNF-α [227].

On the other hand, it has been shown that the presence of bacterial molecules such as LPS or vitamins can also influence the biological properties of MSCs. It has been seen that treatment of PDLSC with IFN-γ increases IDO expression at the protein level; this effect is greater when a TLR2 agonist is added, while no such increase is observed with a TLR4 agonist [228]. Likewise, PDLSCs stimulated with TNF-α, IL-1β, or IFN-γ in the presence of vitamin D have been shown to have a greater capacity to decrease CD4^+^ T lymphocyte proliferation than those stimulated only with the pro-inflammatory cytokines [229]. Taken together, these data indicate that the balance between different cytokines plays an important role in the induction and maintenance of the immunoregulatory properties of MSCs, which is also influenced by the presence of other molecules.

It is currently proposed that MSCs release EVs, which are able to establish contact with immune cells and affect their proliferation, differentiation, and effector function. Interestingly, it has been observed that pro-inflammatory cytokines also modify the content of EVs released by MSCS. In this regard, it has been seen that GMSCs treated with TNF-α release exosomes enriched in CD73, which promote the polarization of macrophages towards an M2 phenotype, which correlates with reduced bone loss in a murine model of periodontitis [158]. Subsequently, the same working group demonstrated that treatment of GMSCs with TNF-α or IFN-γ, individually or in combination, does not significantly modify the CD73 expression of MSCs. However, their analysis of EVs showed a significant increase in CD73 and CD5L transport, and they determined that TNF-α and IFN-γ participate synergistically in that effect. Similarly, these EVs also favor M2-type macrophage differentiation [230]. Another conditioning method used is hypoxia, and a study carried out with MSCs derived from dental pulp shows that the release of EVs is increased in DPSCs subjected to hypoxic conditions, which has a greater capacity to favor M2 macrophage polarization, suppress osteoclast formation, and decrease bone loss in a murine model of LPS-induced inflammatory calvarial bone [231].

Due to the above, it has been proposed that EVs, by carrying immunoregulatory molecules and having an effect similar to that observed with MSCs, could be used as an alternative or complementary therapy to the use of whole cells; the possibility of using them as vehicles for drugs directed against transformed cells has also been proposed. In this regard, it has been seen that EVs released by GMSCs exposed to Paclitaxel contain the drug and are able to decrease the proliferation of transformed cell lines, including pancreatic cancer cells and squamous cell carcinoma [232]. Similar results have been observed with DPSCs which, when cultured with gemcitabine, release EVs loaded with the chemotherapeutic, which decreases the in vitro proliferation of pancreatic carcinoma cell lines [233].

## 6. D-MSCs in Reported Clinical Trials

The analysis of most published research reviewed in the present work shows that D-MSCs have been used for tissue engineering studies in large animals to assess their potential in preclinical applications. The results highlight their potential in dental medicine to regenerate pulp, bone, and periodontal tissues. In this regard, using natural and synthetic scaffolds with D-MSCs to regenerate entire teeth is also an active research area. Although numerous breakthroughs have been made in stem cell research, their success and applicability in clinical trials still need to be determined. Furthermore, D-MSCs possess immunomodulatory properties and have been used in preclinical studies and clinical trials for various diseases.

Due to the properties of D-MSCs, some working groups have begun to use them in clinical trials as treatments for various diseases, most of which involve accessing the oral cavity. To date, there are at least 26 published clinical trials in which D-MSCs are used as treatments (Table 3). The sources used in these tests were DPSCs, SHED, PDLSCs, and SCAPs. Most of the samples used were autologous, but some trials included allogeneic samples. Most of the studies were carried out to support the use of D-MSCs in promoting regeneration in the maxillofacial region. In evaluating the efficacy of D-MSCs for cell-based therapies, the clinical trials focused on numerous variables, including mineralization, bone regeneration, and repair, as well as periodontal parameters, such as tooth mobility, probing depth, gingival recession, the level of clinical insertion, and testing for vital pulp. Therefore, most clinical trials have evaluated the regeneration of tooth-supporting tissues and related parameters. The D-MSCs were administered intravenously, in suspension, via cell sheets, or, for the most part, via scaffolds made mainly of natural polymers, such as collagen. 

Interestingly, although D-MSCs have been shown to interact effectively with immune cells in vitro and in animal models, clinical trials that have explored their immunomodulatory actions are scarce.

The first reported clinical trial was carried out by d’Aquino and collaborators in 2009. In this work, DPSCs embedded in a collagen sponge were used to repair bone defects generated after the extraction of third molars. The results revealed complete bone tissue regeneration and that the cortical level was higher in the experimental group than in the control group after three months of surgery. Additionally, the restoration of the periodontal tissue behind the second molars was also achieved with the proposed intervention. In addition, the patients did not experience symptoms of graft rejection or other complications, suggesting that DPSCs could be used safely and effectively in reconstructive and orthopedic surgery [234].

The second trial used periodontal ligament progenitors for bone regeneration in patients with periodontitis. The results revealed the regeneration of periodontal tissue, and the clinical evaluations demonstrated decreased tooth movement, a decrease in probing depth and attachment level, an increase in gingival resection, and a gain in attachment. In addition, no patients had inflammation or other complications, suggesting that the use of this cell source was safe [235]. In 2014, Shiehzadeh and collaborators used SHED and SCAPs with an injectable poly(lactide-coglycolide)-polyethylene glycol scaffold delivery system in the treatment of large periapical lesions. After a period of 4 months, both sources regenerated the apices of the root and the adjacent bone tissue from treated patients safely and without complications, confirming that these two sources are safe for use in cell therapy [239].

DPSCs are the most commonly used source in regenerative treatments. In addition to being used after the extraction of third molars [234,246,259], they have been applied to increase the floor of the maxillary sinus [236], to regenerate bone defects caused by periodontitis [237,245,247], osteoradionecrosis [238], and intrabony defects [240], mandibular bone defects caused by ameloblastomas [242] and irreversible pulpitis [243,251,258], and to regenerate teeth compromised by periodontal disease [248]. In the majority of these trials, bone regeneration was observed, the probing depth was decreased, and gingival resection was improved. In cases of pulpitis, the pulp tissue was regenerated and was functional. Finally, in the case of bone deformations due to ameloblastoma, a highly invasive and recurrent odontogenic tumor, in addition to successful bone regeneration, there was no recurrence of the tumor after DPSC transplantation.

PDLSCs have mainly been used in the treatment of bone defects caused by periodontitis [241,250,253] and have similar results to those observed with DPSCs. SHED are the second most widely used source of D-MSCs. They have been used to treat apical lesions [244,252], pulp necrosis [249], and malformations present in the palate and cleft lip [254]. These cells can generate bone tissue and functional pulp tissue. On the other hand, SHED have been used in patients suffering from type 2 diabetes mellitus, and although they cause some negative symptoms, such as fever, they are not serious; moreover, they improve glucose metabolism and the function of the pancreatic islets, suggesting that their use is safe and effective [256]. 

A Brazilian company evaluated the safety and tolerability of an intravenous injection of allogenic cell therapy manufactured from human SHED (Nestacell HD, Cellavita, Brazil) in patients with Huntington’s disease. Preliminary findings on the efficacy of this approach have also been assessed, providing initial evidence of allogenic SHED-based cell therapy in the treatment of this debilitating neurological disorder. A case report derived for this clinical trial evaluated whether SHED could migrate to tumors. These cells were administered intravenously to a patient with a lung nodule that was later confirmed to be an adenocarcinoma. The results showed that these cells did not graft (homing) at this point, so they did not generate conditions that favored the growth of the tumor, suggesting that these cells can be used as a treatment for other conditions even when tumors are present [257]. Finally, SHED-CMs have been used to treat erectile dysfunction, and the results showed that they improved erections in treated patients without causing adverse effects, suggesting that the use of these CMs is also safe and effective [255].

In addition to these clinical trials, three protocols have been reported for their dissemination and use in the clinic. The authors proposed the use of DPSCs as a treatment for COVID-19 [260] and cerebrovascular accidents [261,262]. All of these reports indicate that D-MSCs are safe and effective in clinical use; however, it is necessary to increase the understanding and use of these cells in other clinical scenarios that also urgently require effective treatments, such as immunological diseases. Although the immunomodulatory mechanisms of these cells have not yet been fully discerned, preclinical studies have shown promising results. 

For example, cell therapy research for treating oral mucosal disorders such as oral submucous fibrosis, chronic oral ulcers, and mucositis due to chemo- or radiation therapy is limited to animal models. These conditions contain an immunological/inflammatory component, and the present-day treatment modalities consist of steroids and antioxidants, which provide only short-term and symptomatic relief and leave the patient with a certain amount of morbidity. Therefore, human clinical trials on oral mucosal disorders and D-MSCs’ immunological mechanisms are urgently required.

Some important points regarding the existent clinical trials are that only one tooth must be extracted to expand and thus obtain the number of cells needed for the treatments, and autologous samples can be used even when patients have inflammatory conditions, such as periodontitis, since the cells do not lose their proliferative and regenerative properties; these are negative points that occur in BM-MSCs [236,246,250,258,261]. 

One point that has not been considered relevant in clinical trials but that is convenient to emphasize is the effect of age on D-MSCs. It is known that the biological properties of BM-MSCs and other sources decrease when they are obtained from older donors [263,264,265]. In the case of DPSCs, it has been observed that cells from young and elderly donors have similar inhibitory properties on Th 1 and 2 lymphocytes and the secretion of IFN-γ and IL-4, however other characteristics are modified. For example, DPSCs derived from older adults promote the differentiation of Tregs and induce the secretion of TGF-β while DPSCs from young adults promote the differentiation of Th17 and the secretion of IL-6, IL-17a, and HGF [266]. Similar results have been reported in GMSCs. MSCs from elderly donors decrease the proliferation of MNCs and the secretion of inflammatory cytokines in a similar way to MSCs from young donors, concluding that the immunomodulatory potential is not affected by age; however, other biological properties such as differentiation capacity may be negatively altered by this parameter [267]. 

On the other hand, PDLSCs from elderly people do considerably decrease their biological properties, including their immunomodulatory activity [268]. In DFSCs, it has been observed in vitro that the senescence of these cells decreases their biological properties such as proliferation [269]; although it is not known whether obtaining them from elderly donors reduces the characteristics of this population, it has been observed that the total tissue suffers alterations in its metabolic pathways and has a greater risk of inflammatory infiltrate in older donors [270,271], which suggests the possibility that DPSCs could lose their anti-inflammatory potential. For this reason, it is important to further investigate the age-related effects of different sources of D-MSCs to determine which source is the most suitable for use in these clinical trials with minimal risks.

## 7. D-MSCs in Registered Clinical Trials

Currently, at least twenty-three clinical trials are registered on ClinicalTrials.gov (https://clinicaltrials.gov/; the URL was accessed on 28 December 2023), and three more are registered on the International Clinical Trials Registry Platform (ICTRP, https://trialsearch.who.int/; the URL was accessed on 28 December 2023) without reporting results. These protocols were registered between 2010 and 2023 (Table 4).

The predominant sources of D-MSCs are DPSCs, mentioned in thirteen trials: four treatments investigated GMSCs, six investigated SHED, two investigated PDLSCs, and one investigated MSCs obtained from oral mucosa, and one trial described the use of SCAPs. Several of these trials are in phase 1 or phase 1–2, with cells being administered via scaffolds or intravenously. Some trials aimed to administer different concentrations of cells to determine the maximum number of cells that patients could tolerate. These doses ranged from 1 × 10^6^ to 3 × 10^8^, and one or more doses were applied. Sometimes, depending on the protocol, several new parameters needed to be evaluated, such as the levels of inflammatory cytokines, but this depended largely on the protocol to which the use of these D-MSCs was directed. Patients were followed up for between 3 months and 1 year after the transplant.

Most of these protocols focus on dental applications, such as periodontitis, bone defects, gingival defects, and pulp necrosis. However, different research groups have aimed to evaluate their effects on other conditions, such as depression, osteoarthritis, Huntington’s disease, liver cirrhosis, and COVID-19. Although D-MSCs have proven to be safe and effective in clinical treatments carried out to date, it is crucial to continue studying their capabilities in regenerative therapy both at a clinical and preclinical level, establishing more strict and specific criteria in order to reduce potential risks and allow us to control them. However, it is necessary to begin developing more clinical protocols that focus on the other properties of these cells, such as anti-inflammatory potential. To date, preclinical studies have demonstrated the efficacy and safety of the use of these cells in some models of immunological disease, which will possibly begin to venture into the development of protocols against these diseases in humans in the near future.

The advantage of using D-MSCs in clinical trials is that tissue sources, such as the human dental pulp, gingiva, or alveolar bone, are tissues usually discarded in routine dental procedures and can be obtained without significant ethical concerns. GMSCs are also the most favorable source derived from dental tissues because of their high proliferation potential and plasticity; they are easily accessible, expandable in vitro, and exhibit immunomodulatory properties, making them effective in stem cell-based therapy. However, the analysis presented here reveals that DPSCs are the most commonly used MSCs derived from dental tissue in preclinical and clinical trials, including bone, vascular, and neural regeneration applications. There is limited information comparing the immunomodulatory properties of D-MSCs with other sources, and it is evident that further research is necessary to gain a deeper understanding of their properties and potential uses in modulating the immune response.

The heterogeneity of dental tissues has restricted the clinical application of D-MSCs; however, this complication can be addressed by controlling the relevant surface markers and epigenetic modifications. Also, a comprehensive comparative profile of D-MSCs is necessary to improve clinical decisions. Moreover, an ongoing limitation highlighted by most reviewed clinical trials is that many injectable cells are required for D-MSCs cell-based therapies, and the technical simplification of procedures is necessary to improve clinical feasibility.

## 8. Conclusions

The findings from the publications analyzed in this study suggest that BM-MSCs, AT-MSCs, and D-MSCs have similar properties in terms of proliferation, migration, and the ability to differentiate into three distinct types of cells: osteoblasts, chondrocytes, and adipocytes. However, BM-MSC and D-MSC are the most suitable sources for regenerating mineralized tissues in the oral region. 

Clinical trials have shown that treatments with D-MSCs are effective. In addition to continuing to study their clinical safety for tissue regeneration, it is important to study other relevant aspects, such as their immunomodulatory properties, which are fundamental for the development of alternative treatments for autoimmune diseases and other inflammatory conditions. In addition, multiple biophysical cues determine the functions and biology of D-MSCs. For example, D-MSCs cultivated on certain matrices or scaffolds display characteristics that are absent in their monolayer counterparts. Hence, one way to enhance their therapeutic potential is to implant specialized constructs comprising D-MSC-laden scaffolds or hydrogels. In this regard, the use of preclinical information on this topic is encouraging. However, further exploration of the mechanisms involved in this biological property is still necessary. An important and little-explored aspect of MSCs derived from the oral cavity is the effect of an inflammatory environment on their immunomodulatory function, which is relevant for their possible clinical applications. 

Another important limitation is the lack of knowledge we have about other factors that can modify the properties of these cells, such as age, which has been observed to affect the biological properties of some sources more prominently than others. These drawbacks lead us to a third limitation of the use of D-MSCs. Are there complications from using allogeneic D-MSCs samples? Although it is known that MSCs have characteristics that grant them certain immunoprivileges and therefore the risks of using allogeneic samples instead of autologous samples are considerably reduced, this must be studied in D-MSCs in greater depth to determine that the risks of using allogeneic samples are minimal.

## Figures and Tables

**Figure 1 ijms-25-01986-f001:**
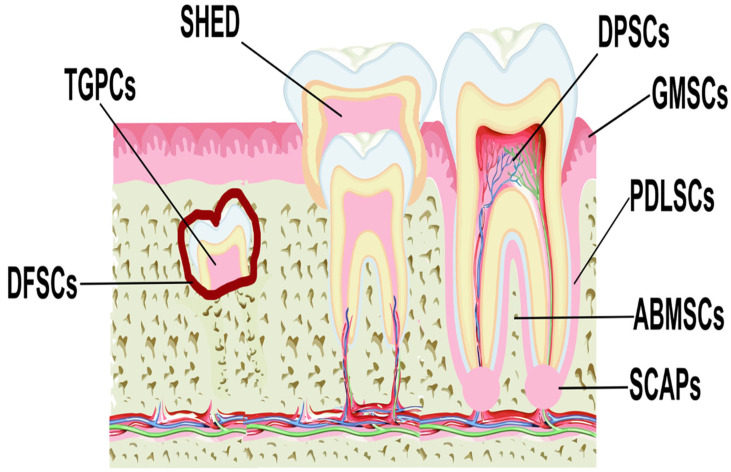
Locations of different sources of MSCs derived from dental tissues. Abbreviations: DPSCs, dental pulp stem cells; PDLSCs, periodontal ligament stem cells; GMSCs, gingiva-derived MSCs; SCAPs, stem cells from the apical papilla; DFSCs, dental follicle stem cells; SHED, stem cells from exfoliated deciduous teeth; ABMSCs, alveolar bone-derived MSCs; TGPCs, tooth germ progenitor cells.

**Figure 2 ijms-25-01986-f002:**
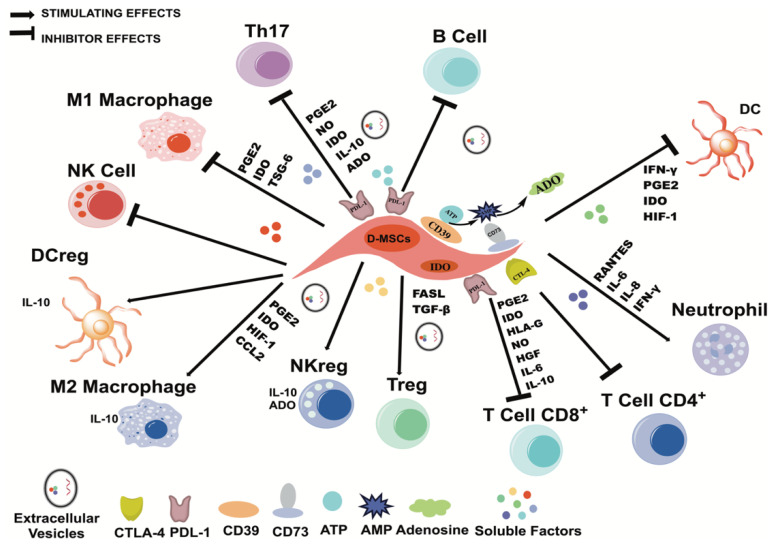
Immunomodulatory properties of MSCs derived from dental tissues. D-MSCs modulate the immune system through different mechanisms, inhibiting proliferation, differentiation, maturation, inflammatory cytokine production, and cytotoxicity. In addition, they can induce the generation of regulatory cells that promote an anti-inflammatory response. D-MSCs can be secreted through soluble factors such as IL-10, PGE2, and TGF-β, the presence of intra- and extracellular enzymes such as IDO, CD39, and CD73, which produce anti-inflammatory molecules, the expression of membrane molecules such as PDL-1, and the secretion of EVs that carry anti-inflammatory molecules.

**Table 1 ijms-25-01986-t001:** Characteristics of D-MSCs.

Source	Efficiency of Isolation	Surface Markers	Embryonic Markers	Neural Markers	Differentiation Potential
DPSCs	++++	CD13, CD29, CD44, CD59, CD73, CD90, CD105, CD146	STRO-1, OCT-4, Nanog, SSEA-1, SEEA-4, SOX-2	β3-tubulin, NFM, Nestin, CNPase, S100, CD271	Adipogenic, osteogenic, odontoblast, angiogenic, and neuronal cells
PDLSCs	++++	CD10, CD29, CD44, CD73, CD105	SSEA-1, SSEA-3, SSEA-4, TRA-1–60, TRA-1–81, OCT-4, Nanog, SOX-2, REX1, ALP	Nestin, OCT-4, SSEA-4, CD271, SOX-10	Adipogenic, chondrogenic, osteogenic, and neuronal cells
GMSCs	++++	CD73, CD90, CD105	SSEA-4, OCT-4, Nanog	Nestin, SOX10, β3-tubulin, NFM, CNPase	Adipogenic, chondrogenic, osteogenic, angiogenic, and neuronal cells
SCAPs	+++	CD24, CD44, CD90, CD146, STRO-1	OCT-4, Nanog, NOTCH-1, SOX-2	OCT-4, SOX2, Nestin	Adipogenic, chondrogenic osteogenic, odontogenic, and neuronal cells
DFSCs	++	CD13, CD29, CD44, CD56, CD59, CD90, CD105, CD106, CD166, STRO-1	OCT-4, Nanog, NOTCH-1, SOX-2	OCT-4, SOX2, Nestin	Osteogenic, odontogenic, and cementogenic
SHED	+++	CD29, CD73, CD90, CD146, CD166	OCT-4, Nanog, SSEA-3, SSEA-4, NOTCH-1, SOX-2	β3-tubulin, NFM, Nestin, CNPase, GAD, NeuN, GFAP, CD271, Vimentin, OCT-4, PAX-6, NSE, MAP-2, PSA- NCAM	Adipogenic, chondrogenic, osteogenic, odontogenic, angiogenic, and neuronal cells
ABMSCs	++++	CD73, CD90, CD105, STRO-1	Oct4, KLF4, Sox2, cMyc	NF-M, NeuN, GFAP	Adipogenic, chondrogenic, and osteogenic
TGPCs	+	CD29, CD73, CD90, CD105, CD166	Nanog, OCT-4, SOX-2, Klf4, c-Myc	Nestin	Adipogenic, chondrogenic, osteogenic, and neuronal cells

Reviewed in [69,70,71,72,73,74,75,76,77,78].

**Table 2 ijms-25-01986-t002:** Molecules involved in the immunoregulation of MSCs from different sources.

Source	Molecules and Mechanisms Related to Immunomodulatory Effects	References
BM-MSCs	IDO, TGF-β1, HGF, IL-6, IL-10, HLA-G, PGE2, NO, ICAM-1, PDL-1 and 2, TSG-6, galectins, EVs	[86,89,99,113,116,121,122,123,124,125,126,127,128]
UCB-MSCs	IDO, TGF-β, HGF, HLA-G, PDL-1, PGE2, galectins	[129,130,131,132]
AT-MSCs	TGF-β, HGF, PGE2, IDO, PDL-1, IL-10, TSG-6, EVs	[97,98,133,134,135,136,137]
DPSCs	IL-6, TGF-β, PGE2, IDO, HLAG, HGF, HIF-1, PDL-1, Fas-FasL pathway, osteoprotegerin, EVs, ADO	[60,114,138,139,140,141,142,143,144]
PDLSCs	TGF-β, IDO, HGF, PGE2, RANTES, eotaxina, IFN-γ, induced protein 10, MCP-1, IL-6, IL-8, IL-1ra EVs, PD-1, PDL-1 and 2	[145,146,147,148,149]
GMSCs	Fas-FasL pathway, PGE2, IDO, iNOS, IL-10, adenosinergic pathway (CD39, CD73, ADO), EVs	[42,59,110,111,150,151,152,153,154,155,156,157,158,159]
SCAPs	IL-6, IL-10, PD-1, PDL-1 y 2 TGF-β 1 y 2, galectin 1, PGE2, EVs	[160,161,162,163]
DFSCs	TGF-β, IDO, TGF-β3, thrombospondin 1, IL-4	[164,165,166]
SHED	TGF-β, IL-10, PDL-1, EVs, Fas-FasL pathway	[167,168,169,170]
ABMSCs	IL-6, MCP-1, PGE2, TIMP-1 and 2, osteoprotegerin	[171,172]
TGPCs	Not described	-

**Table 3 ijms-25-01986-t003:** Published clinical trials carried out with D-MSCs.

Disorder	MSCs Source	Administration Route	Outcomes	Reference
Mandibular bone defects after the extraction of third molars	Autologous DPSCs	Implanted at the extraction sites in a collagen sponge	Increased mineralization rate, cortical bone levels, BMP-2, and VEGF	[234]
Periodontal bone defects	Autologous periodontal ligament progenitors	Surgery, CALCITITE 4060-2 complex bone graft material	Improves gingival recession and decreases probing depth	[235]
Sinus lifting	Autologous DPSCs	Micrografts in the nasal sinus on a collagen sponge	Bone density of the nasal sinus increased	[236]
Periodontal bone defects	Autologous DPSCs	Surgery, on a collagen sponge	Bone tissue regeneration	[237]
Osteoradionecrosis	Autologous DPSCs	Grafted into the affected area, cells soaked in tricalcium phosphate	Regeneration of mandibular bone tissue	[238]
Periradicular periodontitis	Autologous SHED and SCAPs	Implanted in a polyethylene glycol polylactic–polyglycolic acid scaffold	Closure of the apex and bone regeneration	[239]
Intrabony defects	Autologous DPSCs	Surgery, on a collagen sponge	Probing depth decreased and bone tissue increased	[240]
Periodontal bone defects	Autologous PDLSCs	Surgery, on bovine bone materials	Regeneration of alveolar bone tissue	[241]
Mandibular defect due to ameloblastoma	Autologous DPSCs	Packed inside a mesh and placed over the mandible	Mandibular bone regeneration	[242]
Irreversible pulpitis	Autologous mobilized DPSCs	Injected with atelocollagen and G-CSF	They regenerated the pulp and formed dentin	[243]
Periapical lesions	Allogeneic SHED	Administered in suspension in the root canal	Closure of the apex, healing of the periapical tissue, and regeneration of the pulp	[244]
Chronic periodontitis	Autologous DPSCs	Grafted on a collagen sponge	Probing depth decreased and bone tissue increased	[245]
Mandibular bone defects after extraction of third molars	Autologous DPSCs	Implanted at extraction sites on collagen scaffolds	No effect was observed	[246]
Chronic periodontitis	Autologous DPSCs	Micrografts in a collagen sponge	Reduction in probing depth and bone regeneration	[247]
Periodontal disease	Allogeneic DPSCs	Surgery, implantation in a lyophilized collagen–polyvinylpyrrolidone sponge	Probing depth was reduced and bone tissue increased	[248]
Pulp necrosis	Autologous SHED	Implantation of the cells in the affected teeth	Pulp tissue regeneration	[249]
Periodontitis	Autologous PDLSCs	Cellular sheets	Probing depth was reduced and bone tissue was increased	[250]
Irreversible pulpitis	Autologous DPSCs	Administered at the affected site	Regeneration of functional pulp tissue	[251]
Apical lesions	Autologous SHED	Administration in root canals	Healing of the periapical tissue and the formation of functional pulp tissue	[252]
Intrabony lesions	Autologous PDLSCs	Surgery, implantation in the affected sites	Reduction in probing depth	[253]
Cleft lip and palate	Autologous SHED	Surgery, implantation in a hydroxyapatite–collagen sponge	Generation of bone tissue	[254]
Erectile dysfunction	CMs SHED	Injections into the corpora cavernosa of the penis	Improvement of erections	[255]
Diabetes mellitus 2	Allogeneic SHED	Intravenous	Decreased glycosylated hemoglobin and glycosylated serum albumin levels and caused fever, fatigue, and rash	[256]
Huntington’s disease associated with primary lung adenocarcinoma	Allogeneic SHED	Intravenous	Cells did not graft (homing) in the pre-existing tumor	[257]
Irreversible pulpitis	Autologous DPSCs	Administration of the cells in the root canals combined with G-CSF in atelocollagen	Regeneration of functional pulp tissue	[258]
Mandibular bone defects after extraction of third molars	Autologous DPSCs	Implanted in the affected areas	Probing depth was reduced	[259]

**Table 4 ijms-25-01986-t004:** Registered clinical trials using D-MSCs.

ID	Study Status	Conditions	Interventions	Phases	Study Type
NCT04302519	Unknown	COVID-19	DPSCs	Early Phase 1	Interventional
NCT04983225	Active not recruiting	Periodontitis	DPSCs	Phase 1	Interventional
NCT02731586	Unknown	Edentulous alveolar ridge	DPSCs	Early Phase 1	Interventional
NCT05924373	Not recruiting	Periodontitis	DPSCs	Phase 2	Interventional
NCT05127369	Not recruiting	Depression	DPSCs	NA	Interventional
NCT02728115	Active not recruiting	Huntington’s disease	DPSCs	Phase 1	Interventional
NCT04130100	Unknown	Knee osteoarthritis	DPSCs	Early Phase 1	Interventional
NCT03570333	Active not recruiting	Gingival disorders	GMSCs	Na	Interventional
NCT02728115	Active not recruiting	Huntington’s disease	SHED (Cellavita HD)	Phase 1	Interventional
NCT04219241	Active not recruiting	Huntington’s disease	SHED (Cellavita HD)	Phase 2|Phase 3	Interventional
NCT03252535	Completed	Huntington’s disease	SHED (Cellavita HD)	Phase 2	Interventional
NCT01082822	Unknown	Periodontitis	PDLSCs	Phase 1|Phase 2	Interventional
NCT04336254	Unknown	COVID-19	DPSCs	Phase 1|Phase 2	Interventional
NCT02209311	Unknown	Alveolar bone atrophy	MSCs from oral mucosa	Phase 1|Phase 2	Interventional
NCT06043453	Recruiting	Apical periodontitis	DPSCs and SCAPs	NA	Observational
NCT03957655	Unknown	Liver cirrhosis	SHED	Early Phase 1	Interventional
NCT01814436	Unknown	Dental pulp necrosis	SHED	NA	Interventional
NCT05728346	Not recruiting	Dental pulp necrosis	SHED	NA	Interventional
NCT03638154	Completed	Periodontal defects	GMSCs	NA	Interventional
NCT03137979	Unknown	Periodontitis	GMSCs	Phase 1|Phase 2	Interventional
NCT04434794	Completed	Gingival recession	GMSCs	Phase 1|Phase 2	Interventional
NCT02464202	Completed	Tooth transplantation	PDLSCs	NA	Interventional
NCT02523651	Unknown	Periodontal diseases	DPSCs	Phase 1|Phase 2	Interventional
ChiCTR2300073144	Not recruiting	Periodontitis	DPSCs	Phase 2	Interventional
RBR-65trt53	Not recruiting	COVID-19	DPSCs	Phase 1|Phase 2	Interventional
IRCT20140911019125N8	Recruiting	COVID-19	DPSCs	Phase 2|Phase 3	Interventional

## Data Availability

Not applicable.

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
