# Peer review of "Mesenchymal Stromal Cells Derived from Dental Tissues: Immunomodulatory Properties and Clinical Potential"

_ijms, 2024, doi:10.3390/ijms25041986_

Round 1
Reviewer 1 Report
Comments and Suggestions for Authors
1、 It is recommended to add a summarizing paragraph to each section of the manuscript, which will be more helpful for the reader to understand it in more depth, due to the fact that the review should contain the author's own views and opinions.
2、 The structure of the manuscript seems to be confusing. If D-MSCs include the dental pulp (DPSCs), periodontal ligament (PDLSCs), gingival tissue (GMSCs), apical papilla (SCAPs), dental follicle (DFPCs) , human exfoliated deciduous teeth (SHEDs) , alveolar bone-derived MSCs (ABMSCs), and tooth germ progenitor cells (TGPCs), then the subsequent 5-10 paragraphs should not be juxtaposed with the previous paragraphs.
3、 Could the subtitle of the author's paragraph be more specific, such as "Immunomodulatory properties of D-MSCs". Are the following paragraphs 5-10 also about "Immunomodulatory properties"?
4、 The purpose and research significance of this paper should be clearly stated in the abstract and Introduction of the manuscript. A brief description in Introduction is recommended. In addition, the content of the Abstract should be consistent with the final conclusions.
Reviewer 2 Report
Comments and Suggestions for Authors
In this review, the immunoregulatory mechanisms identified at the preclinical level in combination with different types of MSCs found in dental tissues are described, as well as a description of clinical trials in which MSCs from these sources have been used.
1. Line 33 needs a reference, the following reference may be useful: DOI: 10.2174/1574888X17666211220100521
2. 432 lines do not have references.
3. Which mesenchymal stromal cells derived from dental tissue have more advantages than others? Which mesenchymal stromal cells derived from dental tissue are the most used in preclinical and clinical studies?
4. The limitations of using any mesenchymal stromal cell derived from dental tissue should be written.
Reviewer 3 Report
Comments and Suggestions for Authors
Round 2
Reviewer 1 Report
Comments and Suggestions for Authors
I recommend accepting the manuscript.